# CAN RECURRENT NEURAL NETWORKS WARP TIME?

**Corentin Tallec**
Laboratoire de Recherche en Informatique
Université Paris Sud
Gif-sur-Yvette, 91190, France
corentin.tallec@u-psud.fr

**Yann Ollivier**
Facebook Articial Intelligence Research
Paris, France
yol@fb.com

## ABSTRACT

Successful recurrent models such as long short-term memories (LSTMs) and gated recurrent units (GRUs) use *ad hoc* gating mechanisms. Empirically these models have been found to improve the learning of medium to long term temporal dependencies and to help with vanishing gradient issues.

We prove that learnable gates in a recurrent model formally provide *quasi-invariance to general time transformations* in the input data. We recover part of the LSTM architecture from a simple axiomatic approach.

This result leads to a new way of initializing gate biases in LSTMs and GRUs. Experimentally, this new *chrono initialization* is shown to greatly improve learning of long term dependencies, with minimal implementation effort.

Recurrent neural networks (e.g. (Jaeger, 2002)) are a standard machine learning tool to model and represent temporal data; mathematically they amount to learning the parameters of a parameterized dynamical system so that its behavior optimizes some criterion, such as the prediction of the next data in a sequence.

Handling long term dependencies in temporal data has been a classical issue in the learning of recurrent networks. Indeed, stability of a dynamical system comes at the price of exponential decay of the gradient signals used for learning, a dilemma known as the *vanishing gradient* problem (Pascanu et al., 2012; Hochreiter, 1991; Bengio et al., 1994). This has led to the introduction of recurrent models specifically engineered to help with such phenomena.

Use of feedback connections (Hochreiter & Schmidhuber, 1997) and control of feedback weights through gating mechanisms (Gers et al., 1999) partly alleviate the vanishing gradient problem. The resulting architectures, namely long short-term memories (LSTMs (Hochreiter & Schmidhuber, 1997; Gers et al., 1999)) and gated recurrent units (GRUs (Chung et al., 2014)) have become a standard for treating sequential data.

Using orthogonal weight matrices is another proposed solution to the vanishing gradient problem, thoroughly studied in (Saxe et al., 2013; Le et al., 2015; Arjovsky et al., 2016; Wisdom et al., 2016; Henaff et al., 2016). This comes with either computational overhead, or limitation in representational power. Furthermore, restricting the weight matrices to the set of orthogonal matrices makes forgetting of useless information difficult.

The contribution of this paper is threefold:

- We show that postulating invariance to time transformations in the data (taking invariance to time warping as an axiom) necessarily leads to a gate-like mechanism in recurrent models (Section 1). This provides a clean derivation of part of the popular LSTM and GRU architectures from first principles. In this framework, gate values appear as time contraction or time dilation coefficients, similar in spirit to the notion of time constant introduced in (Mozer, 1992).

- From these insights, we provide precise prescriptions on how to initialize gate biases (Section 2) depending on the range of time dependencies to be captured. It has previously been advocated that setting the bias of the forget gate of LSTMs to 1 or 2 provides overall good performance (Gers & Schmidhuber, 2000; Jozefowicz et al., 2015). The viewpoint here

explains why this is reasonable in most cases, when facing medium term dependencies, but fails when facing long to very long term dependencies.

- We test the empirical benefits of the new initialization on both synthetic and real world data (Section 3). We observe substantial improvement with long-term dependencies, and slight gains or no change when short-term dependencies dominate.

# 1 FROM TIME WARPING INVARIANCE TO GATING

When tackling sequential learning problems, being resilient to a change in time scale is crucial. Lack of resilience to time rescaling implies that we can make a problem arbitrarily difficult simply by changing the unit of measurement of time. Ordinary recurrent neural networks are highly non-resilient to time rescaling: a task can be rendered impossible for an ordinary recurrent neural network to learn, simply by inserting a fixed, small number of zeros or whitespaces between all elements of the input sequence. An explanation is that, with a given number of recurrent units, the class of functions representable by an ordinary recurrent network is not invariant to time rescaling.

Ideally, one would like a recurrent model to be able to learn from time-warped input data $x(c(t))$ as easily as it learns from data $x(t)$, at least if the time warping $c(t)$ is not overly complex. The change of time $c$ may represent not only time rescalings, but, for instance, accelerations or decelerations of the phenomena in the input data.

We call a class of models *invariant to time warping*, if for any model in the class with input data $x(t)$, and for any time warping $c(t)$, there is another (or the same) model in the class that behaves on data $x(c(t))$ in the same way the original model behaves on $x(t)$. (In practice, this will only be possible if the warping $c$ is not too complex.) We will show that this is deeply linked to having gating mechanisms in the model.

**Invariance to time rescaling**
Let us first discuss the simpler case of a linear time rescaling. Formally, this is a linear transformation of time, that is

$$c \quad : \quad \begin{array}{ccc} \mathbb{R}_+ & \longrightarrow & \mathbb{R}_+ \\ t & \longmapsto & \alpha t \end{array} \tag{1}$$

with $\alpha > 0$. For instance, receiving a new input character every 10 time steps only, would correspond to $\alpha = 0.1$.

Studying time transformations is easier in the continuous-time setting. The discrete time equation of a basic recurrent network with hidden state $h_t$,

$$h_{t+1} = \tanh\left(W_x\, x_t + W_h\, h_t + b\right) \tag{2}$$

can be seen as a time-discretized version of the continuous-time equation[1]

$$\frac{\mathrm{d}h(t)}{\mathrm{d}t} = \tanh\left(W_x\, x(t) + W_h\, h(t) + b\right) - h(t) \tag{3}$$

namely, (2) is the Taylor expansion $h(t + \delta t) \approx h(t) + \delta t \frac{\mathrm{d}h(t)}{\mathrm{d}t}$ with discretization step $\delta t = 1$.

Now imagine that we want to describe time-rescaled data $x(\alpha t)$ with a model from the same class. Substituting $t \leftarrow c(t) = \alpha t$, $x(t) \leftarrow x(\alpha t)$ and $h(t) \leftarrow h(\alpha t)$ and rewriting (3) in terms of the new variables, the time-rescaled model satisfies[2]

$$\frac{\mathrm{d}h(t)}{\mathrm{d}t} = \alpha\, \tanh\left(W_x\, x(t) + W_h\, h(t) + b\right) - \alpha\, h(t). \tag{4}$$

However, when translated back to a discrete-time model, this no longer describes an ordinary RNN but a *leaky* RNN (Jaeger, 2002, §8.1). Indeed, taking the Taylor expansion of $h(t + \delta t)$ with $\delta t = 1$ in (4) yields the recurrent model

$$h_{t+1} = \alpha\, \tanh\left(W_x\, x_t + W_h\, h_t + b\right) + (1 - \alpha)h_t \tag{5}$$

---

[1] We will use indices $h_t$ for discrete time and brackets $h(t)$ for continuous time.

[2] More precisely, introduce a new time variable $T$ and set the model and data with variable $T$ to $H(T) := h(c(T))$ and $X(T) := x(c(T))$. Then compute $\frac{\mathrm{d}H(T)}{\mathrm{d}T}$. Then rename $H$ to $h$, $X$ to $x$ and $T$ to $t$ to match the original notation.

Thus, a straightforward way to ensure that a class of (continuous-time) models is able to represent input data $x(\alpha t)$ in the same way that it can represent input data $x(t)$, is to take a leaky model in which $\alpha > 0$ is a learnable parameter, corresponding to the coefficient of the time rescaling. Namely, the class of ordinary recurrent networks is not invariant to time rescaling, while the class of leaky RNNs (5) is.

Learning $\alpha$ amounts to learning the global characteristic timescale of the problem at hand. More precisely, $1/\alpha$ ought to be interpreted as the characteristic forgetting time of the neural network.[3]

**Invariance to time warpings**
In all generality, we would like recurrent networks to be resilient not only to time rescaling, but to all sorts of time transformations of the inputs, such as variable accelerations or decelerations.

An eligible time transformation, or *time warping*, is any increasing differentiable function $c$ from $\mathbb{R}_+$ to $\mathbb{R}_+$. This amounts to facing input data $x(c(t))$ instead of $x(t)$. Applying a time warping $t \leftarrow c(t)$ to the model and data in equation (3) and reasoning as above yields

$$\frac{\mathrm{d}h(t)}{\mathrm{d}t} = \frac{\mathrm{d}c(t)}{\mathrm{d}t} \tanh\left(W_x\, x(t) + W_h\, h(t) + b\right) - \frac{\mathrm{d}c(t)}{\mathrm{d}t}\, h(t). \qquad (6)$$

Ideally, one would like a model to be able to learn from input data $x(c(t))$ as easily as it learns from data $x(t)$, at least if the time warping $c(t)$ is not overly complex.

To be invariant to time warpings, a class of (continuous-time) models has to be able to represent Equation (6) for any time warping $c(t)$. Moreover, the time warping is unknown a priori, so would have to be learned.

Ordinary recurrent networks do not constitute a model class that is invariant to time rescalings, as seen above. A fortiori, this model class is not invariant to time warpings either.

For time warping invariance, one has to introduce a *learnable* function $g$ that will represent the derivative[4] of the time warping, $\frac{\mathrm{d}c(t)}{\mathrm{d}t}$ in (6). For instance $g$ may be a recurrent neural network taking the $x$'s as input.[5] Thus we get a class of recurrent networks defined by the equation

$$\frac{\mathrm{d}h(t)}{\mathrm{d}t} = g(t) \tanh\left(W_x\, x(t) + W_h\, h(t) + b\right) - g(t)\, h(t) \qquad (7)$$

where $g$ belongs to a large class (universal approximator) of functions of the inputs.

The class of recurrent models (7) is *quasi*-invariant to time warpings. The quality of the invariance will depend on the learning power of the learnable function $g$: a function $g$ that can represent any function of the data would define a class of recurrent models that is perfectly invariant to time warpings; however, a specific model for $g$ (e.g., neural networks of a given size) can only represent a specific, albeit large, class of time warpings, and so will only provide quasi-invariance.

Heuristically, $g(t)$ acts as a time-dependent version of the fixed $\alpha$ in (4). Just like $1/\alpha$ above, $1/g(t_0)$ represents the local forgetting time of the network at time $t_0$: the network will effectively retain information about the inputs at $t_0$ for a duration of the order of magnitude of $1/g(t_0)$ (assuming $g(t)$ does not change too much around $t_0$).

Let us translate back this equation to the more computationally realistic case of discrete time, using a Taylor expansion with step size $\delta t = 1$, so that $\frac{\mathrm{d}h(t)}{\mathrm{d}t} = \cdots$ becomes $h_{t+1} = h_t + \cdots$. Then the model (7) becomes

$$h_{t+1} = g_t \tanh\left(W_x\, x_t + W_h\, h_t + b\right) + (1 - g_t)\, h_t. \qquad (8)$$

---

[3]Namely, in the "free" regime if inputs stop after a certain time $t_0$, $x(t) = 0$ for $t > t_0$, with $b = 0$ and $W_h = 0$, the solution of (4) is $h(t) = e^{-\alpha\,(t-t_0)}h(t_0)$, and so the network retains information from the past $t < t_0$ during a time proportional to $1/\alpha$.

[4]It is, of course, algebraically equivalent to introduce a function $g$ that learns the derivative of $c$, or to introduce a function $G$ that learns $c$. However, only the derivative of $c$ appears in (6). Therefore the choice to work with $\frac{\mathrm{d}c(t)}{\mathrm{d}t}$ is more convenient. Moreover, it may also make learning easier, because the simplest case of a time warping is a time rescaling, for which $\frac{\mathrm{d}c(t)}{\mathrm{d}t} = \alpha$ is a constant. Time warpings $c$ are increasing by definition: this translates as $g > 0$.

[5]The time warping has to be learned only based on the data seen so far.

where $g_t$ itself is a function of the inputs.

This model is the simplest extension of the RNN model that provides invariance to time warpings.[6] It is a basic gated recurrent network, with input gating $g_t$ and forget gating $(1 - g_t)$.

Here $g_t$ has to be able to learn an arbitrary function of the past inputs $x$; for instance, take for $g_t$ the output of a recurrent network with hidden state $h^g$:

$$g_t = \sigma(W_{gx}\, x_t + W_{gh}\, h_t^g + b_g) \tag{9}$$

with sigmoid activation function $\sigma$ (more on the choice of sigmoid below). Current architectures just reuse for $h^g$ the states $h$ of the main network (or, equivalently, relabel $h \leftarrow (h, h^g)$ to be the union of both recurrent networks and do not make the distinction).

The model (8) provides invariance to *global* time warpings, making all units face the same dilation/contraction of time. One might, instead, endow every unit $i$ with its own local contraction/dilation function $g^i$. This offers more flexibility (gates have been introduced for several reasons beyond time warpings (Hochreiter, 1991)), especially if several unknown timescales coexist in the signal: for instance, in a multilayer model, each layer may have its own characteristic timescales corresponding to different levels of abstraction from the signal. This yields a model

$$h_{t+1}^i = g_t^i \tanh\left(W_x^i\, x_t + W_h^i\, h_t + b^i\right) + (1 - g_t^i)\, h_t^i \tag{10}$$

with $h^i$ and $(W_x^i, W_h^i, b^i)$ being respectively the activation and the incoming parameters of unit $i$, and with each $g^i$ a function of both inputs and units.

Equation 10 defines a simple form of gated recurrent network, that closely resembles the evolution equation of *cell* units in LSTMs, and of hidden units in GRUs.

In (10), the forget gate is tied to the input gate ($g_t^i$ and $1 - g_t^i$). Such a setting has been successfully used before (e.g. (Lample et al., 2016)) and saves some parameters, but we are not aware of systematic comparisons. Below, we *initialize* LSTMs this way but do not enforce the constraint throughout training.

**Continuous time versus discrete time**
Of course, the analogy between continuous and discrete time breaks down if the Taylor expansion is not valid. The Taylor expansion is valid when the derivative of the time warping is not too large, say, when $\alpha \lesssim 1$ or $g_t \lesssim 1$ (then (8) and (7) are close). Intuitively, for continuous-time data, the physical time increment corresponding to each time step $t \to t + 1$ of the discrete-time recurrent model should be smaller than the speed at which the data changes, otherwise the situation is hopeless. So discrete-time gated models are invariant to time warpings that stretch time (such as interspersing the data with blanks or having long-term dependencies), but obviously not to those that make things happen too fast for the model.

Besides, since time warpings are monotonous, we have $\frac{\mathrm{d}c(t)}{\mathrm{d}t} > 0$, i.e., $g_t > 0$. The two constraints $g_t > 0$ and $g_t < 1$ square nicely with the use of a sigmoid for the gate function $g$.

## 2   TIME WARPINGS AND GATE INITIALIZATION

If we happen to know that the sequential data we are facing have temporal dependencies in an approximate range $[T_{\min}, T_{\max}]$, it seems reasonable to use a model with memory (forgetting time) lying approximately in the same temporal range. As mentioned in Section 1, this amounts to having values of $g$ in the range $\left[\frac{1}{T_{\max}}, \frac{1}{T_{\min}}\right]$.

The biases $b_g$ of the gates $g$ greatly impact the order of magnitude of the values of $g(t)$ over time. If the values of both inputs and hidden layers are centered over time, $g(t)$ will typically take values

---

[6]Even more: the weights $(W_x, W_h, b)$ are the same for $h(t)$ in (3) and $h(c(t))$ in (6). This means that *in principle it is not necessary to re-train the model for the time-warped data*. (Assuming, of course, that $g_t$ can learn the time warping efficiently.) The variable copy task (Section 3) arguably illustrates this. So the definition of time warping invariance could be strengthened to use the *same* model before and after warping.

centered around $\sigma(b_g)$. Values of $\sigma(b_g)$ in the desired range $\left[\frac{1}{T_{\max}}, \frac{1}{T_{\min}}\right]$ are obtained by choosing the biases $b_g$ between $-\log(T_{\max} - 1)$ and $-\log(T_{\min} - 1)$. This is a loose prescription: we only want to control the order of magnitude of the memory range of the neural networks. Furthermore, we don't want to bound $g(t)$ too tightly to some value forever: if rare events occur, abruplty changing the time scale can be useful. Therefore we suggest to use these values as initial values only.

This suggests a practical initialization for the bias of the gates of recurrent networks such as (10): when characteristic timescales of the sequential data at hand are expected to lie between $T_{\min}$ and $T_{\max}$, initialize the biases of $g$ as $-\log(\mathcal{U}([T_{\min}, T_{\max}]) - 1)$ where $\mathcal{U}$ is the uniform distribution[7].

For LSTMs, using a variant of (Graves et al., 2013):

$$i_t = \sigma(W_{xi}\, x_t + W_{hi}\, h_{t-1} + b_i) \tag{11}$$
$$f_t = \sigma(W_{xf}\, x_t + W_{hf}\, h_{t-1} + b_f) \tag{12}$$
$$c_t = f_t\, c_{t-1} + i_t\, \tanh(W_{xc}\, x_t + W_{hc}\, h_{t-1} + b_c) \tag{13}$$
$$o_t = \sigma(W_{xo}\, x_t + W_{ho}\, h_{t-1} + b_o) \tag{14}$$
$$h_t = o_t\, \tanh(c_t), \tag{15}$$

the correspondence between between the gates in (10) and those in (13) is as follows: $1 - g_t$ corresponds to $f_t$, and $g_t$ to $i_t$. To obtain a time range around $T$ for unit $i$, we must both ensure that $f_t^i$ lies around $1 - 1/T$, and that $i_t$ lies around $1/T$. When facing time dependencies with largest time range $T_{\max}$, this suggests to initialize LSTM gate biases to

$$\begin{aligned} b_f &\sim \log(\mathcal{U}([1, T_{\max} - 1])) \\ b_i &= -b_f \end{aligned} \tag{16}$$

with $\mathcal{U}$ the uniform distribution and $T_{\max}$ the expected range of long-term dependencies to be captured.

Hereafter, we refer to this as the *chrono initialization*.

## 3 EXPERIMENTS

First, we test the theoretical arguments by explicitly introducing random time warpings in some data, and comparing the robustness of gated and ungated architectures.

Next, the chrono LSTM initialization is tested against the standard initialization on a variety of both synthetic and real world problems. It heavily outperforms standard LSTM initialization on all synthetic tasks, and outperforms or competes with it on real world problems.

The synthetic tasks are taken from previous test suites for RNNs, specifically designed to test the efficiency of learning when faced with long term dependencies (Hochreiter & Schmidhuber, 1997; Le et al., 2015; Graves et al., 2014; Martens & Sutskever, 2011; Arjovsky et al., 2016).

In addition (Appendix A), we test the chrono initialization on next character prediction on the Text8 (Mahoney, 2011) dataset, and on next word prediction on the Penn Treebank dataset (Mikolov et al., 2012). Single layer LSTMs with various layer sizes are used for all experiments, except for the word level prediction, where we use the best model from (Zilly et al., 2016), a 10 layer deep recurrent highway network (RHN).

**Pure warpings and paddings.** To test the theoretical relationship between gating and robustness to time warpings, various recurrent architectures are compared on a task where the only challenge comes from warping.

The unwarped task is simple: remember the previous character of a random sequence of characters. Without time warping or padding, this is an extremely easy task and all recurrent architectures are successful. The only difficulty will come from warping; this way, we explicitly test the robustness of various architectures to time warping and nothing else.

---

[7]When the characteristic timescales of the sequential data at hand are completetly unknown, a possibility is to draw, for each gate, a random time range $T$ according to some probability distribution on $\mathbb{N}$ with slow decay (such as $\mathbb{P}(T = k) \propto \frac{1}{k \log (k+1)^2}$) and initialize biases to $\log(T)$.

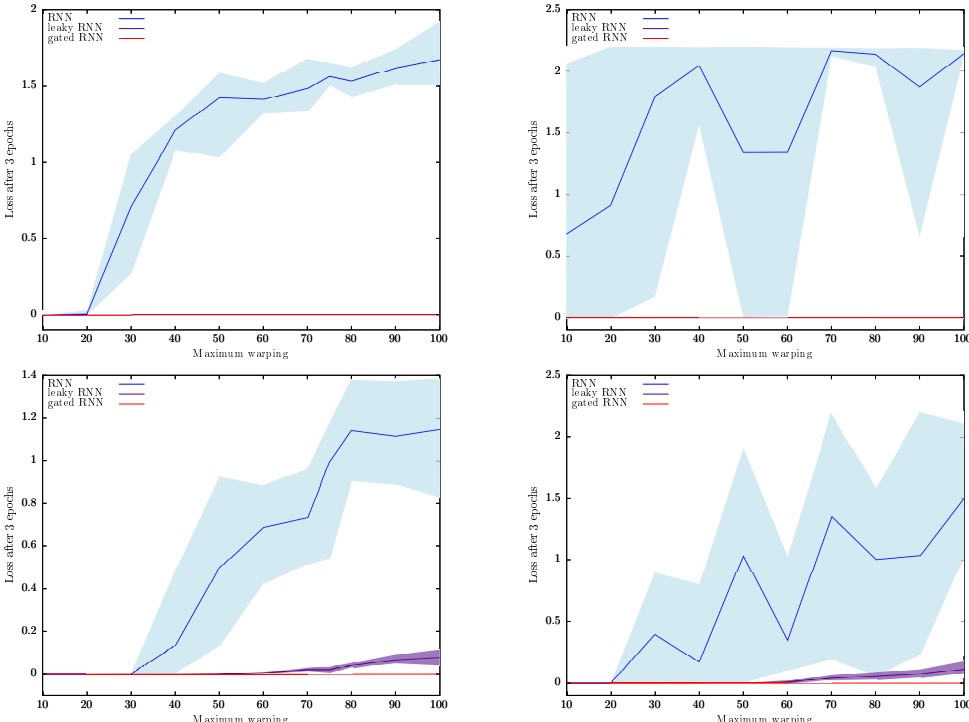

Figure 1: Performance of different recurrent architectures on warped and padded sequences sequences. From top left to bottom right: uniform time warping of length `maximum_warping`, uniform padding of length `maximum_warping`, variable time warping and variable time padding, from 1 to `maximum_warping`. (For uniform padding/warpings, the leaky RNN and gated RNN curves overlap, with loss 0.) Lower is better.

Unwarped task example:

Input:     `All human beings are born free and equal`
Output:    `All human beings are born free and equa`

Uniform warping example (warping $\times 4$):

Input:    `AAAAllllllll     hhhhuuuummmmaaaannnn`
Output:   `AAAAllllllll     hhhhuuuummmmaaaa`

Variable warping example (random warping $\times 1$–$\times 4$):

Input:    `Allllll   hhhummmmaannn bbbbeeiiingssss`
Output:  `AAAlllll    huuuummaaan      bbeeeeingggg`

Figure 2: A task involving pure warping.

Uniformly time-warped tasks are produced by repeating each character `maximum_warping` times both in the input and output sequence, for some fixed number `maximum_warping`.

Variably time-warped tasks are produced similarly, but each character is repeated a random number of times uniformly drawn between 1 and `maximum_warping`. The same warping is used for the input and output sequence (so that the desired output is indeed a function of the input). This exactly corresponds to transforming input $x(t)$ into $x(c(t))$ with $c$ a random, piecewise affine time warping. Fig. 2 gives an illustration.

For each value of `maximum_warping`, the train dataset consists of $50,000$ length-500 randomly warped random sequences, with either uniform or variable time warpings. The alphabet is of size 10 (including a dummy symbol). Contiguous characters are enforced to be different. After warping, each sequence is truncated to length 500. Test datasets of $10,000$ sequences are generated similarly. The criterion to be minimized is the cross entropy in predicting the next character of the output sequence.

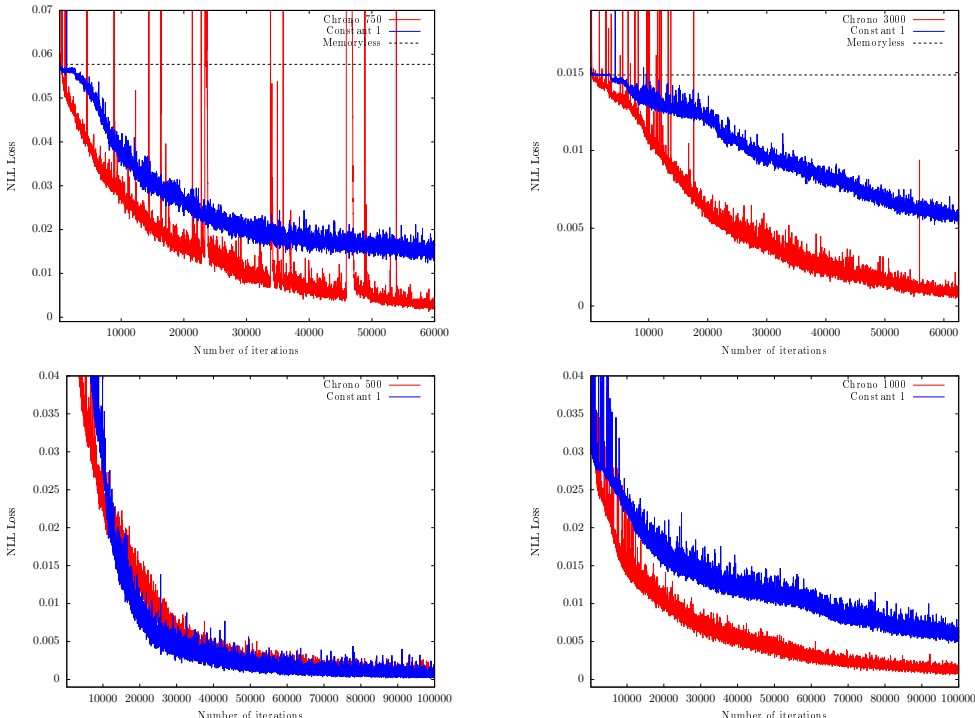

Figure 3: Standard initialization (blue) vs. chrono initialization (red) on the copy and variable copy task. From left to right, top to bottom, standard copy $T = 500$ and $T = 2000$, variable copy $T = 500$ and $T = 1000$. Chrono initialization heavily outperforms standard initialization, except for variable length copy with the smaller $T$ where both perform well.

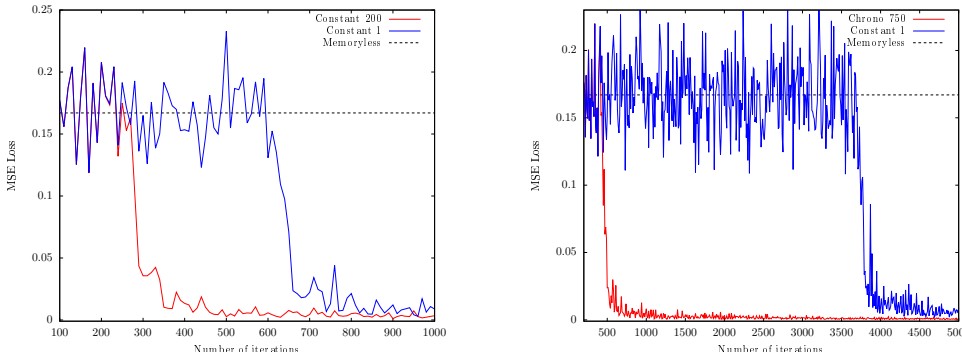

Figure 4: Standard initialization (blue) vs. chrono initialization (red) on the adding task. From left to right, $T = 200$, and $T = 750$. Chrono initialization heavily outperforms standard initialization.

Note that each sample in the dataset uses a new random sequence from a fixed alphabet, and (for variable warpings) a new random warping.

A similar, slightly more difficult task uses *padded* sequences instead of warped sequences, obtained by padding each element in the input sequence with a fixed or variable number of 0's (in continuous-time, this amounts to a time warping of a continuous-time input sequence that is nonzero at certain points in time only). Each time the input is nonzero, the network has to output the previous nonzero character seen.

We compare three recurrent architectures: RNNs (Eq. (2), a simple, ungated recurrent network), leaky RNNs (Eq. (5), where each unit has a constant learnable "gate" between $0$ and $1$) and gated RNNs, with one gate per unit, described by (10). All networks contain $64$ recurrent units.

The point of using gated RNNs (10) ("LSTM-lite" with tied input and forget gates), rather than full LSTMs, is to explicitly test the relevance of the arguments in Section 1 for time warpings. Indeed these LSTM-lite already exhibit perfect robustness to warpings in these tasks.

RMSprop with an $\alpha$ parameter of 0.9 and a batch size of 32 is used. For faster convergence, learning rates are divided by 2 each time the evaluation loss has not decreased after 100 batches. All architectures are trained for 3 full passes through the dataset, and their evaluation losses are compared. Each setup is run 5 times, and mean, maximum and minimum results among the five trials are reported. Results on the test set are summarized in Fig. 1.

Gated architectures significantly outperform RNNs as soon as moderate warping coefficients are involved. As expected from theory, leaky RNNs perfectly solve uniform time warpings, but fail to achieve optimal behavior with variable warpings, to which they are not invariant. Gated RNNs, which are quasi invariant to general time warpings, achieve perfect performance in both setups for all values of `maximum_warping`.

**Synthetic tasks.** For synthetic tasks, optimization is performed using RMSprop (Tieleman & Hinton, 2012) with a learning rate of $10^{-3}$ and a moving average parameter of 0.9. No gradient clipping is performed; this results in a few short-lived spikes in the plots below, which do not affect final performance.

COPY TASKS. The copy task checks whether a model is able to remember information for arbitrarily long durations. We use the setup from (Hochreiter & Schmidhuber, 1997; Arjovsky et al., 2016), which we summarize here. Consider an alphabet of 10 characters. The ninth character is a dummy character and the tenth character is a signal character. For a given $T$, input sequences consist of $T + 20$ characters. The first 10 characters are drawn uniformly randomly from the first 8 letters of the alphabet. These first characters are followed by $T - 1$ dummy characters, a *signal* character, whose aim is to signal the network that it has to provide its outputs, and the last 10 characters are dummy characters. The target sequence consists of $T + 10$ dummy characters, followed by the first 10 characters of the input. This dataset is thus about remembering an input sequence for exactly $T$ timesteps. We also provide results for the *variable* copy task setup presented in (Henaff et al., 2016), where the number of characters between the end of the sequence to copy and the signal character is drawn at random between 1 and $T$.

The best that a memoryless model can do on the copy task is to predict at random from among possible characters, yielding a loss of $\frac{10 \log(8)}{T+20}$ (Arjovsky et al., 2016).

On those tasks we use LSTMs with 128 units. For the standard initialization (baseline), the forget gate biases are set to 1. For the new initialization, the forget gate and input gate biases are chosen according to the chrono initialization (16), with $T_{\max} = \frac{3T}{2}$ for the copy task, thus a bit larger than input length, and $T_{\max} = T$ for the variable copy task. The results are provided in Figure 3.

Importantly, our LSTM baseline (with standard initialization) already performs better than the LSTM baseline of (Arjovsky et al., 2016), which did not outperform random prediction. This is presumably due to slightly larger network size, increased training time, and our using the bias initialization from (Gers & Schmidhuber, 2000).

On the copy task, for all the selected $T$'s, chrono initialization largely outperforms the standard initialization. Notably, it does not plateau at the memoryless optimum. On the variable copy task, chrono initialization is even with standard initialization for $T = 500$, but largely outperforms it for $T = 1000$.

ADDING TASK. The adding task also follows a setup from (Hochreiter & Schmidhuber, 1997; Arjovsky et al., 2016). Each training example consists of two input sequences of length $T$. The first one is a sequence of numbers drawn from $\mathcal{U}([0, 1])$, the second is a sequence containing zeros everywhere, except for two locations, one in the first half and another in the second half of the sequence. The target is a single number, which is the sum of the numbers contained in the first sequence at the positions marked in the second sequence.

The best a memoryless model can do on this task is to predict the mean of $2 \times \mathcal{U}([0, 1])$, namely 1 (Arjovsky et al., 2016). Such a model reaches a mean squared error of 0.167.

LSTMs with $128$ hidden units are used. The baseline (standard initialization) initializes the forget biases to $1$. The chrono initialization uses $T_{\max} = T$. Results are provided in Figure 4. For all $T$'s, chrono initialization significantly speeds up learning. Notably it converges 7 times faster for $T = 750$.

## CONCLUSION

The self loop feedback gating mechanism of recurrent networks has been derived from first principles via a postulate of invariance to time warpings. Gated connections appear to regulate the local time constants in recurrent models. With this in mind, the chrono initialization, a principled way of initializing gate biases in LSTMs, has been introduced. Experimentally, chrono initialization is shown to bring notable benefits when facing long term dependencies.

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

## A   ADDITIONAL EXPERIMENTS

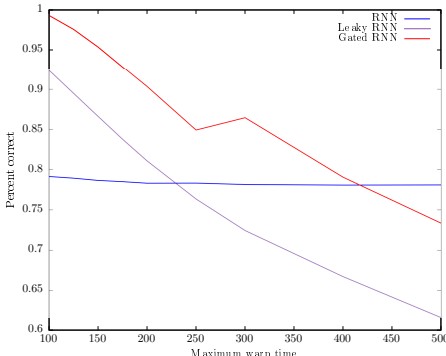

Figure 5: Generalization performances of different recurrent architectures on the warping problem. Networks are trained with uniform warps between 1 and 50 and evaluated on uniform warps between 100 and a variable maximum warp.

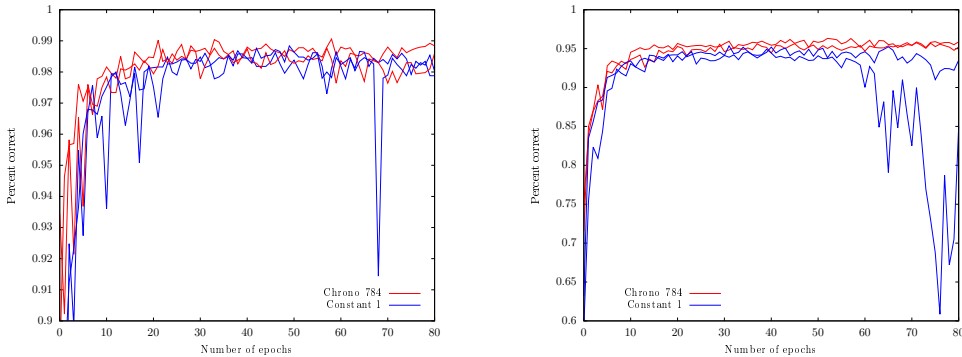

Figure 6: Standard initialization (blue) vs. chrono initialization (red) on pixel level classification tasks. From left to right, MNIST and pMNIST.

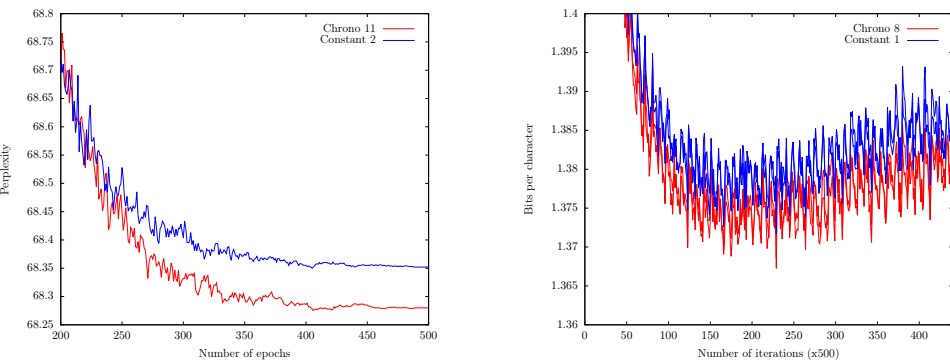

Figure 7: Standard initialization (blue) vs. chrono initialization (red) on the word level PTB (left) and on the character level text8 (right) validation sets.

**On the generalization capacity of recurrent architectures.**   We proceeded to test the generalization properties of RNNs, leaky RNNs and chrono RNNs on the pure warping experiments presented in Section 3. For each of the architectures, a recurrent network with 64 recurrent units is trained for 3 epochs on a variable warping task with warps between 1 and 50. Each network is then tested on

warped sequences, with warps between $100$ and an increasingly big maximum warping. Results are summarized in Figure 5.

All networks display reasonably good, but not perfect, generalization. Even with warps 10 times longer than the training set warps, the networks still have decent accuracy, decreasing from $100\%$ to around $75\%$.

Interestingly, plain RNNs and gated RNNs display a different pattern: overall, gated RNNs perform better but their generalization performance decreases faster with warps eight to ten times longer than those seen during training, while plain RNN never have perfect accuracy, below $80\%$ even within the training set range, but have a flatter performance when going beyond the training set warp range.

**Pixel level classification: MNIST and pMNIST.** This task, introduced in (Le et al., 2015), consists in classifying images using a recurrent model. The model is fed pixels one by one, from top to bottom, left to right, and has to output a probability distribution for the class of the object in the image.

We evaluate standard and chrono initialization on two image datasets: MNIST (LeCun et al., 1999) and permuted MNIST, that is, MNIST where all images have undergone the same pixel permutation.

LSTMs with $512$ hidden units are used. Once again, standard initialization sets forget biases to $1$, and the chrono initialization parameter is set to the length of the input sequences, $T_{\max} = 784$. Results on the validation set are provided in Figure 6. On non-permuted MNIST, there is no clear difference, even though the best validation error is obtained with chrono initialization. On permuted MNIST, chrono initialization performs better, with a best validation result of $96.3\%$, while standard initialization obtains a best validation result of $95.4\%$.

**Next character prediction on text8.** Chrono initialization is benchmarked against standard initialization on the character level text8 dataset (Mahoney, 2011). Text8 is a 100M character formatted text sample from Wikipedia. (Mikolov et al., 2012)'s train-valid-test split is used: the first 90M characters are used as training set, the next 5M as validation set and the last 5M as test set.

The exact same setup as in (Cooijmans et al., 2016) is used, with the code directly taken from there. Namely: LSTMs with $2000$ units, trained with Adam (Kingma & Ba, 2014) with learning rate $10^{-3}$, batches of size $128$ made of non-overlapping sequences of length $180$, and gradient clipping at $1.0$. Weights are orthogonally initialized, and recurrent batch normalization (Cooijmans et al., 2016) is used.

Chrono initialization with $T_{\max} = 8$ is compared to standard $b_f = 1$ initialization. Results are presented in Figure 7. On the validation set, chrono initialization uniformly outperforms standard initialization by a small margin. On the test set, the compression rate is $1.37$ with chrono initialization, versus $1.38$ for standard initialization.[8] This same slight difference is observed on two independent runs.

Our guess is that, on next character prediction, with moderately sized networks, short term dependencies dominate, making the difference between standard and chrono initialization relatively small.

**Next word prediction on Penn Treebank.** To attest for the resilience of chrono initialization to more complex models than simple LSTMs, we train on word level Penn Treebank (Mikolov et al., 2012) using the best deep RHN network from (Zilly et al., 2016). All hyperparameters are taken from of (Zilly et al., 2016). For the chrono bias initialization, a single bias vector $b$ is sampled according to $b \sim \log(\mathcal{U}(1, T_{\max}))$, the carry gate bias vectors of all layers are initialized to $-b$, and the transform gate biases to $b$. $T_{\max}$ is chosen to be $11$ (because this gives an average bias initialization close to the value 2 from (Zilly et al., 2016)).[9] Without further hyperparameter search and with a single run, we obtain test results similar to (Zilly et al., 2016), with a test perplexity of $6.54$.

---

[8]Both those results are slightly below the $1.36$ reported in (Cooijmans et al., 2016), though we use the same code and same random seed. This might be related to a smaller number of runs, or to a different version of the libraries used.

[9]This results in small characteristic times: RHNs stack $D$ update steps in every timestep, where $D$ is the depth of the RHN, so timescales are divided by $D$.

