# OpenReview forum: "Can recurrent neural networks warp time?"
_ICLR.cc/2018/Conference — Accept (Poster)_

### Official Review · AnonReviewer1 · 2017-11-27
**Cool theoretical contribution with rather unrelated experiments**

**Rating:** 8
**Confidence:** 4

**Review:**

tl;dr:
 - The paper has a really cool theoretical contribution.
 - The experiments do not directly test whether the theoretical insight holds in practice, but instead a derivate method is tested on various benchmarks.

I must say that this paper has cleared up quite a few things for me. I have always been a skeptic wrt LSTM, since I myself did not fully understand when to prefer them over vanilla RNNs for reasons other than “they empirically work much better in many domains.” and “they are less prone to vanishing gradients”.

Section 1 is a bliss: it provides a very useful candidate explanation under which conditions vanilla RNNs fail (or at least, do not efficiently generalise) in contrast to gated cells. I am sincerely happy about the write up and will point many people to it.

The major problem with the paper, in my eyes, is the lack of experiments specific to test the hypothesis. Obviously, quite a bit of effort has gone into the experimental section. The focus however is comparison to the state of the art in terms of raw performance.

That leaves me asking: are gated RNNs superior to vanilla RNNs if the data is warped?
Well, I don’t know now. I only can say that there is reason to believe so.

I *really* do encourage the authors to go back to the experiments and see if they can come up with an experiment to test the main hypothesis of the paper. E.g. one could make synthetic warpings, apply it to any data set and test if things work out as expected. Such a result would in my opinion be of much more use than the tiny increment in performance that is the main output of the paper as of now, and which will be stomped by some other trick in the months to come. It would be a shame if such a nice theoretical insight got under the carpet because of that. E.g. today we hold [Pascanu 2013] dear not because of the proposed method, but because of the theoretical analysis.

Some minor points.
- The authors could make use of less footnotes, and try to incorporate them into the text or appendix.
- A table of results would be nice.
- Some choices of the experimental section seem arbitrary, e.g. the use of optimiser and to not use clipping of gradients. In general, the evaluation of the hyper parameters is not rigorous.
- “abruplty” -> “abruptly” on page 5, 2nd paragraph

### References
[Pascanu 2013] Pascanu, Razvan, Tomas Mikolov, and Yoshua Bengio. "On the difficulty of training recurrent neural networks." International Conference on Machine Learning. 2013.

---

> ### Author Response · Authors · 2018-01-02
> **Answer to reviewer 3**
>
> Thank you for your insightful review.  The lack of an experiment to test the main theoretical result of the paper was indeed a major flaw of the original version. We have updated the paper accordingly, by adding experiments with pure warpings/pure paddings to the main track of the paper (pages 6-8), and moving less central experiments to the appendix. We hope this experiment is more or less what you had in mind. The results are in line with the theoretical derivations: plain RNNs cannot account for warpings, leaky RNNs can
> account for uniform time scalings but not irregular warpings, and gated RNNs can adapt to irregular warpings.

---

### Official Review · AnonReviewer3 · 2017-11-28
**Interesting development to gated RNN**

**Rating:** 8
**Confidence:** 4

**Review:**

The paper provides an interesting theoretical explanation on why gated RNN architectures such as LSTM and GRU work well in practice. The paper shows how "gate values appear as time contraction or time dilation coefficients". The authors also point out the connection between the gate biases and the range of time dependencies captured in the network. From that, they develop a simple yet effective initialization method which performs well on different datasets.

Pros:
- The idea is interesting, it well explain the success of gated RNNs.
- Writing: The paper is well written and easy to read.

Cons:
Experiments: only small datasets were used in the experiments, it would be more convincing if the author could use larger datasets. One suggestion to make the experiment more complete is to gradually increase the initial value of the biases to see how it affect the performance. To use 'chrono initialization', one need to estimate the range of time dependency which could be difficult in practice.

---

> ### Author Response · Authors · 2018-01-02
> **Answer to Reviewer 2**
>
> Thank you for your constructive comments! As for larger-scale
> experiments, we had to make choices and decided to focus on the setup suggested by Reviewer 3, namely, to center the experiments specifically on the theoretical claims of Section 1. This is now included in the text, while MNIST, pMNIST and next-step prediction have been moved to the appendix instead. We are aware this is not what you suggested, but as pointed out by Reviewer 3, we think this is more in line with the central claim of this work.

---

### Official Review · AnonReviewer2 · 2017-11-28
**A simple and important insight about LSTMs**

**Rating:** 8
**Confidence:** 4

**Review:**

Summary:
This paper shows that incorporating invariance to time transformations in recurrent networks naturally results in a gating mechanism used by LSTMs and their variants. This is then used to develop a simple bias initialization scheme for the gates when the range of temporal dependencies relevant for a problem can be estimated or are known. Experiments demonstrate that the proposed initialization speeds up learning on synthetic tasks, although benefits for next-step prediction tasks are limited.

Quality and significance:
The core insight of the paper is the link between recurrent network design and its effect on how the network reacts to time transformations. This insight is simple, elegant and valuable in my opinion.

It is becoming increasingly apparent recently that the benefits of the gating and cell mechanisms introduced by the LSTM, now also used in feedforward networks, go beyond avoiding vanishing gradients. The particular structural elements also induce certain inductive biases which make learning or generalization easier in many cases. Understanding the link between model architecture and behavior is very useful for the field in general, and this paper contributes to this knowledge. In light of this, I think it is reasonable to ignore the fact that the proposed initialization does not provide benefits on Penn Treebank and text8. The real value of the paper is in providing an alternative way of thinking about LSTMs that is theoretically sound and intuitive.

Clarity:
The paper is well-written in general and easy to understand. A minor complaint is that there are an unnecessarily large number of paragraph breaks, especially on pages 3 and 4, which make reading slightly jarring.

---

> ### Author Response · Authors · 2018-01-02
> **Answer to Reviewer 1**
>
> Thank you for the constructive review of our paper. We've noted your
> remarks on the limited benefits of the method for next-step prediction,
> along with those of reviewer 3. Consequently, we have revised the paper to include an experiment on pure warping/padding that specifically tests the main theoretical claim made in the paper. Accordingly, MNIST, pMNIST and next-step prediction have been moved to the appendix instead.

---

> > ### Comment · AnonReviewer2 · 2018-01-09
> > **Questions**
> >
> > I'm happy to see the added experiments. One question: In Fig. 1, are the reported losses computed on the test set? In any case, I think it would be interesting to include results for both training and test losses for these experiments.
> >
> > Another question: what happens if the networks are tested on a test set with maximum_warping higher than that used during training?

---

> > > ### Author Response · Authors · 2018-01-10
> > > **Answers**
> > >
> > > Losses are computed on the evaluation set for all setups. We chose not to plot both the train and evaluation curves as both exhibit very similar trends. The difference between the architectures is not an effect of overfitting.
> > >
> > > For generalizing to longer warpings than those of the training set, all networks display reasonably good (but not perfect) generalization. Even with warps 10 times longer than the training set warps, the networks still have decent accuracy, decreasing from 100% to around 75%. We tested this today, by training the architectures with max_warp=50, and evaluating on sequences with warps from 100 to 500.
> > >
> > > Interestingly, plain RNNs and gated RNNs display a different pattern: overall, gated RNNs perform better, but their generalization performance decreases faster with warps 8x-10x max_warp, while plain RNN never have perfect accuracy (always below 80% even within the training set range) but have a flatter performance when going beyond the training set max_warp; the two curves cross at about 9x max_warp, at 75%-80% accuracy.

---

> > > > ### Comment · AnonReviewer2 · 2018-01-12
> > > > **Reply**
> > > >
> > > > Thanks for the answers. These results are very interesting and important.
> > > >
> > > > I urge the authors to include this extended analysis with any other discussions (of plots etc.) in the supplementary material. Since they have already done these experiments, it will be helpful for any future readers.
> > > >
> > > > I am willing to upgrade my rating if the authors add this analysis.

---

> > > > > ### Author Response · Authors · 2018-01-15
> > > > > **Added analysis in the supplementary**
> > > > >
> > > > > The generalization analysis, along with a curve showing precision vs warp range, with warps longer than those seen during training was added. We hope this is what you had in mind.

---

### Author Response · Authors · 2018-01-03
**Addition of experiments validating the core hypothesis**

In response to the reviewers insightful comments, two experiments precisely testing the invariance properties of simple recurrent, leaky and gated networks have been added, which validate the theoretical claim.

---

### Author Response · Authors · 2018-01-15
**Generalization analysis**

A generalization analysis, along with a curve showing precision vs warp range, with warps longer than those seen during training was added to the appendix.

---

### Decision · Program_Chairs · 2018-01-29
**ICLR 2018 Conference Acceptance Decision**

**Decision:**

Accept (Poster)

**Comment:**

All the reviews like the theoretical result presented in the paper which relates the gating mechanism of LSTMS (and GRUs) to time invariance / warping. The theoretical result is great and is used to propose a heuristic for setting biases when time invariance scales are known. The experiments are not mind-boggling, but none of the reviewers seem to think that's a show stopper.